# Piperidine Derivatives: Recent Advances in Synthesis and Pharmacological Applications

**DOI:** 10.3390/ijms24032937

**Published:** 2023-02-02

**Authors:** Nikita A. Frolov, Anatoly N. Vereshchagin

**Affiliations:** N. D. Zelinsky Institute of Organic Chemistry, Russian Academy of Sciences, Leninsky Prospect 47, 119991 Moscow, Russia

**Keywords:** piperidines, piperidinones, hydrogenation, cyclization, cycloaddition, annulation, amination, multicomponent reactions, biological activity, pharmacological activity

## Abstract

Piperidines are among the most important synthetic fragments for designing drugs and play a significant role in the pharmaceutical industry. Their derivatives are present in more than twenty classes of pharmaceuticals, as well as alkaloids. The current review summarizes recent scientific literature on intra- and intermolecular reactions leading to the formation of various piperidine derivatives: substituted piperidines, spiropiperidines, condensed piperidines, and piperidinones. Moreover, the pharmaceutical applications of synthetic and natural piperidines were covered, as well as the latest scientific advances in the discovery and biological evaluation of potential drugs containing piperidine moiety. This review is designed to help both novice researchers taking their first steps in this field and experienced scientists looking for suitable substrates for the synthesis of biologically active piperidines.

## 1. Introduction

Piperidine is a six-membered heterocycle including one nitrogen atom and five carbon atoms in the sp3-hybridized state. Piperidine-containing compounds represent one of the most important synthetic medicinal blocks for drugs construction, and their synthesis has long been widespread [1]. Today, it can be unequivocally stated that heterocyclic compounds play a significant part in the pharmaceutical industry, and one of the most common in their structure is the piperidine cycle. Thus, more than 7000 piperidine-related papers were published during the last five years according to Sci-Finder. Therefore, the development of fast and cost-effective methods for the synthesis of substituted piperidines is an important task of modern organic chemistry.

In the last several years, a lot of reviews concerning specific methods of pipiridine synthesis [2,3,4,5,6,7,8,9,10,11,12,13], functionalization [14], and their pharmacological application [15,16,17,18,19,20] were published. Herein, we have summarized the main routes in modern organic chemistry to the synthesis of piperidine derivatives (the scope is displayed in Figure 1, there and further, the atoms and bonds forming the piperidine cycle are highlighted in blue.), as well as their medical applications.

In the first chapter, approaches to the synthesis of piperidines were structured, and the features of the reaction mechanisms, the required conditions, and the limitations and advantages of particular methods were discussed. The second chapter is devoted to the main pharmacological applications of piperidine natural and synthetic derivatives, as well as recent progress in the development of new drugs of this class.

We hope that this review will provide a broad perspective on the field and will attract new creative minds to further develop the piperidine class.

## 2. Recent Advances in the Synthesis of Piperidine Derivatives

In this chapter, recent developments in the field of substituted piperidines synthesis will be discussed. It is worth noting that only examples of the piperidine ring formation will be considered and not the functionalization of already existing ones.

For the purposes of our review, we have distinguished three main routes to the piperidines synthesis (Figure 1).

### 2.1. Hydrogenation/Reduction

Chemists have used hydrogenation reactions since the early 19th century. This fundamental process plays a key role in modern organic synthesis. Over many decades of scientific progress, researchers have developed approaches to the hydrogenation of a wide variety of heterocyclic compounds and their derivatives, including furans [21,22,23], pyrroles [24], indoles [25], thiophenes [26], imidazoles [27], oxazolones [28], quinolines [29], etc. Pyridines are of particular interest for this review, as they are the most common source for obtaining piperidines by this method.

Now, there are many ways to achieve *N*-heteroaromatic compounds hydrogenation. Usually, the reactions take place under transition metal catalysis and harsh conditions (high temperature, great pressure, long reaction time), which makes them more expensive than the use of the other methods. Moreover, in order to meet modern pharmaceutical standards, in most cases, it is necessary to obtain a specific isomer. Thus, the reaction must be stereoselective, which is difficult in view of the aforementioned conditions. However, despite all the obvious problems of this approach, in the last decade, scientists offered various methods for overcoming them.

Here, we have discussed some of the recent developments in this field regarding the preparation of piperidines using metal- and organocatalysis. More information about *N*-heterocyclic hydrogenation methods is represented elsewhere [14,30]. 

In the work of Beller et al., a various pyridine derivatives hydrogenation was discussed [31,32,33]. A new heterogeneous cobalt catalyst based on titanium nanoparticles and melamine allowed for acid-free hydrogenation with good yields and selectivity. It was shown that it is possible to carry out the described conversions of substituted pyridines into the corresponding piperidines in water as a solvent (Figure 2A) [31]. Moreover, the authors have optimized the method for obtaining piperidine-based biologically active substances, including Melperone, a second-generation antipsychotic [34]. Further, the Beller group developed a ruthenium heterogeneous catalyst for the diastereoselective *cis*-hydrogenation of multi-substituted pyridines (Figure 2B) [32] and a previously unknown nickel silicide catalyst (Figure 2C) [33]. It is the first example of a nickel catalyst’s successful application in efficient pyridine hydrogenation. It is worth noting that all catalysts possessed a high stability rate and remained effective after multiple uses.

Along with ruthenium and cobalt, iridium is an effective transition metal for stereoselective catalytic hydrogenation. Thus, Qu et al. reported the successful asymmetric hydrogenation of 2-substituted pyridinium salts using an iridium(I) catalyst containing a *P,N-*ligand (Figure 3) [35]. The authors suggested that the reaction proceeds through the outer-sphere dissociative mechanism, which is known for this type of hydrogenation [36]. The reactant undergoes a series of successive protonations. The product configuration is determined by the stereoselective enamine protonation. This approach is also suitable for high-volume synthesis. Thus, the authors performed the large-scale enantioselective reduction of 2,3-disubstituted indenopyridine as part of the synthesis of a biologically active substance—11β-hydroxysteroid dehydrogenase type 1 inhibitor (11β-HSD1) [37]. 11β-HSD1 is used for treating diseases associated with cortisol abnormalities [38].

Rhodium and palladium are befitting for pyridine hydrogenation as well. For example, in 2019, Glorius et al. developed a strategy for accessing all-*cis*-(multi)fluorinated piperidines from the corresponding fluoropyridines [39,40] (Figure 4). First, the authors used rhodium(I) complex and pinacol borane to achieve highly diastereoselective products through the dearomatization/hydrogenation process (Figure 4A) [39]. As a result, a wide range of substituted fluoropiperidines have been obtained, including fluorinated analogs of commercially available and biologically active substances, including Melperone, Diphenidol, Dyclonine, Eperisone, and Cycrimine. The work represents a major advance in the underdeveloped field of fluoropiperidine derivatives acquirement. However, the method has its limitations in the pyridine moieties range (products with hydroxy, aryl, ester, and amide groups were not affordable) and moisture sensitivity. Therefore, in 2020, the Glorius group came up with another idea for accessing highly valuable fluorinated piperidines. The method is based on palladium-catalyzed hydrogenation (Figure 4B) [40]. The developed approach was suitable for most substrates that were inaccessible by rhodium catalysis and was effective in the presence of air and moisture. It is worth noting that the axial-position for fluorine atoms was prevalent in the majority of experiments.

The interruption of palladium-catalyzed hydrogenation by water led to piperidinones (Figure 5) [41]. The method allows for the furthering of the one-pot functionalization of unsaturated intermediates, which usually requires multiple steps. Moreover, the reaction possessed great selectivity, high yields, and a broad substrate scope.

Grygorenko et al. used palladium and rhodium hydrogenation for their approach to all isomeric (cyclo)alkylpiperidines (Figure 6) [42,43]. The method proposed by the authors combines three reactions in one step: the removal of the metalation group, dehydroxylation, and pyridine reduction (Figure 6A). Unfortunately, due to the acid use, some substrates were not accessible. It is possible to retain the hydroxyl group with relatively higher yields by using triethylamine instead of hydrochloric acid as an additive (Figure 6B) [42]. The rhodium catalyst proved to be more effective when 3-substituted piperidines bearing partially fluorinated groups were synthesized (Figure 6C) [43]. The reaction took place under milder conditions and required significantly less time. Nevertheless, hydrodefluorination might occur in some cases, which leads to an undesirable by-product without any fluorine substituents.

Usuki et al. proposed an interesting example of functionalized chemoselective piperidine synthesis combining multiple stages in one (Figure 7A) [44]. One-pot sequential Suzuki–Miyaura coupling and hydrogenation were carried out under mild conditions. The authors pointed out the utmost importance of maintaining the optimal starting material concentration for the successful hydrogenation process. The mild hydrogenation method was studied in detail on a broad spectrum of substrates (Figure 7B) [45]. The authors concluded that the HOMO/LUMO states and the bulkiness of the substituents majorly influenced the reaction rate.

Moreover, chemoselectivity proved to be useful in the synthesis of donepezil (Figure 7C), a widely used active component for Alzheimer’s disease treatment [46]. Li et al. used a similar approach in the synthesis of alkoxy-piperidine derivatives (Figure 7D) [47]. While the indole moiety remained aromatic, the pyridine part was fully converted into piperidine.

Zhang et al. accomplished a stereoselective coupling/hydrogenation cascade (Figure 8) [48]. After the coupling phase, quaternary pyridinium salt (intermediate) undergoes a partial reduction with Raney-Ni as a catalyst. If sodium tetrahydroborate is used instead of nickel, the reduction goes smoother and tetrahydropyridine is formed. The resulting piperidines can go through further transformations without enantioselectivity loss.

Borenium and hydrosilanes are often used as a non-metal alternative in catalytic hydrogenation [5]. Thus, Crudden et al. discovered that boron ions diastereoselectively reduce substituted pyridines to piperidines in the presence of hydrosilanes (Figure 9A) [49]. For *bis*-substituted pyridines, hydrogenation took place under mild conditions, while for *ortho*-derivatives pressure and temperature were more than doubled. Silanes are necessary to prevent the product-catalyst adduct formation. Wang et al. have developed another method for the hydroboration/hydrogenation cascade of pyridines (Figure 9B) [50]. The method was *cis*-selective and especially effective for 2,3-disubstituted pyridines. The reaction has a number of features. For example, 2-furyl or 2-thienyl substituents undergo ring opening to form alcohols and thiols, respectively. In the reaction of 2,4-substituted pyridines, the reduction was incomplete in certain cases with the formation of tetrahydropyridines. Moreover, it is possible to obtain a cyclic imine with fluorine in the *meta* position.

If hydrogen is not used, then, under similar conditions, as in the abovementioned work of Cruden [49], a dearomative hydrosilylation will occur. Therefore, the resulting enamine can be further functionalized. Joung et al. studied this approach in detail using the example of the hydrosilylation of quinolines (Figure 10A) [51], isoquinolines (Figure 10B), and pyridines (Figure 10C) [52]. Generally, there are two variants of hydrosilylation depending on the position of the substituents in relation to the nitrogen atom. The main route is 1,4-hydrosilylation (Figure 10A,C), when a hydride attack occurs at C4. Whenever *para*-substitution is in place, *N-*heterocycles are reduced in a 1,2-manner (Figure 10B). Chang et al. conducted a thorough mechanistic study [53].

Double reduction is another interesting approach for the asymmetric synthesis of piperidines. Phillips et al. demonstrated the efficient asymmetric synthesis of aminofluoropiperidine as a precursor for the CGRP (calcitonin gene-related peptide receptor) antagonist (Figure 11A) [54].

The first hydrogenation was carried out using sodium tetrahydroborate under mild conditions and with high yields. The key step was the second asymmetric hydrogenation involving a catalytic ruthenium(II) complex. Titanium isopropoxide was used to neutralize the fluorine released during the process, caused by catalyst poisoning. The procedure provides the complete conversion of enamine into the desired piperidine, with a little admixture of the defluorinated by-product (~3%). Qu et al. used a rhodium(I) catalyst with a ferrocene ligand in the similar reaction (Figure 11B) [55]. Under these conditions, the proportion of desfluoro-impurities was less than 1%. Krasavin et al. described the stereoselective hydrogenation of unsaturated substituted piperidinones, followed by the reduction of the lactam group to give *cis*-configured 2,4-disubstituted 1-alkylpiperidines (Figure 11C) [56].

Previously, Li et al. proposed a catalytic complex of rhodium(I) with a *P-*chiral bisphosphorus ligand for the enantioselective asymmetric hydrogenation of aliphatic carbocyclic and heterocyclic tetrasubstituted enamides (Figure 12) [57]. The authors showed that the interaction of the substrate and the ligands isopropyl group could play a significant role in the enantioselectivity. The developed method made it possible to carry out an efficient and practical synthesis of the Janus kinase inhibitor—Tofacitinib.

In conclusion, the synthesis of piperidines by hydrogenation/reduction has been an effective and popular approach in recent years. The main substrates are substituted pyridines with both protected and unprotected nitrogen. Most often, before hydrogenation, the substrate already has all the substituents necessary for the target product. However, current approaches combine hydrogenation and functionalization as a one-pot process, making the synthesis faster and less costly. In addition to the classic metal catalysis, the organocatalysis is gaining popularity.

### 2.2. Intramolecular Cyclization

Intramolecular cyclization (or intramolecular ring closure) is a unique case of an intramolecular reaction in which a cycle is formed within the structure of a singular molecule. Thus, the backbone of the cyclic product is entirely represented in the original reactant.

The initiation of intramolecular cyclization occurs through the activation of various functional groups or bonds. This usually requires the addition of a catalyst, an oxidizing or a reducing agent (depending on the substrate), the maintenance of the environment, etc. The main challenge for this approach is the achievement of stereo- and regioselectivity. Chiral ligands and catalysts can solve this problem. However, the selection of reaction conditions for their stability is a serious obstacle.

In piperidine formation by intramolecular cyclization, the substrate contains a nitrogen source (usually an amino group) and one or more active sites directly involved in cyclization. It is noteworthy that, with the direct participation of the nitrogen atom in cyclization, a new C-N bond is formed. In other cases, the formation of a new C-C bond is observed (Figure 13). In most cases, cyclizations proceeds according to Baldwin’s rules, established by Jack Baldwin in 1976 [58] and revised by Alabugin and Gilmore in 2016 [59]. Possible variants of piperidine cyclization are shown in the figure below (Figure 2).

There are many approaches to the preparation of piperidines by intramolecular ring closure: asymmetric synthesis [60], metal-catalyzed cyclization [9,12], intramolecular silyl-Prins reaction [2,61], electrophilic cyclization [6,13], aza-Michael reaction [8], etc. Further, we outlined the recent updates on this topic within the last five years. For easier understanding, the chapter was divided by substrate classes. Reacting groups and newly formed cycle bonds are highlighted in a red color.

#### 2.2.1. Alkene Cyclization

Nevado et al. proposed a synthetic route for the oxidative amination of non-activated alkenes to form substituted piperidines (Figure 14A) [62]. The reaction is catalyzed by a gold(I) complex and proceeds with the use of the iodine(III) oxidizing agent. The method is designed for the difunctionalization of a double bond with the simultaneous formation of an *N-*heterocycle and the introduction of an *O-*substituent. Liu et al. developed an enantioselective approach to this reaction using a palladium catalyst (previously unknown for that type of amination) with a novel pyridine-oxazoline ligand (Figure 14B) [63]. The authors found that a sterically bulky substituent at the C6 position of the ligand enables the palladium activation of olefins by weakening the PyN−Pd(II) bond, therefore enhancing electrophilicity. This ligand type was also effective in the azidation reaction (Figure 14C) [64]. Moreover, Li et al. developed the ligand-controlled regioselective diamination of alkenes (Figure 14D) [65]. In this case, a more sterically hindered ligand led to pyrrolidone formation. According to the authors, the unusual regioselectivity of the reaction arose because of a significant steric effect of the nucleophilic reagent—*N-*Fluorobenzenesulfonimide (NFSI). Shibata et al. applied palladium catalysis for the intramolecular aminotrifluoromethanesulfinyloxylation of alkenes (Figure 14E) [66]. An intricate complex provided 6-endo-cyclized-type piperidines with moderate yields and diastereoselective ratios. Zawisza et al. represented palladium-catalyzed ligand-free diastereoselective intramolecular allylic amination (Figure 14F) [67]. The protection group with defined stereochemistry played a crucial role in the product selectivity. Thus, the protecting group acts as a chiral ligand. Engle et al. used 8-aminoquinoline as a guide group for the palladium-catalyzed intramolecular hydroamination (Figure 14G) [68]. The reaction is suitable for the synthesis of five and six-membered heterocycles and proceeds via *syn*-addition, a proto-depalladation mechanism. Donohoe et al. presented the stereoselective carboamination of alkenes (Figure 14H) [69]. Cyclization is initiated by the carbocation generated in situ from alcohol. As a result, two new bonds are formed at the same time: C-C (alkylation) and C-N (carboamination).

Intramolecular aza-Michael reactions (IMAMR) are among the most straightforward strategies for constructing plenty of enantiomerically enriched *N-*heterocycles through double bond activation [8,70,71]. Recently, Pozo et al. proposed a general protocol for di- and tri-substituted piperidines synthesis by IMAMR using organocatalysis (Figure 15A,B) [72,73].

The combination of a quinoline organocatalyst and trifluoroacetic acid as a cocatalyst afforded a series of enantiomerically enriched 2,5- and 2,6-disubstituted protected piperidines (Figure 15A) and 2,5,5-trisubstituted protected piperidines (Figure 15B) in good yields. Moreover, the authors found that the ratio of catalysts used plays a key role in the final product isomerization [72]. Bhattacharjee et al. developed the large-scale synthesis of 2,6-*trans*-piperidines through IMAMR, with TBAF as a base (Figure 15C) [74]. For this reaction, cesium carbonate also showed good results (85% yields, *trans*/*cis* = 90/10). However, its use was difficult when scaling up due to its poor solubility. Ye et al. carried out carbene-catalyzed IMAMR (Figure 15D) [75]. The addition of an NHC-catalyst made it possible to achieve good enantioselectivity and higher yields compared with base-only reaction.

Sutherland et al. performed a novel acid-mediated stereoselective intramolecular 6-endo-trig cyclization of enones [76] (Figure 16). When studying the reaction mechanism, it was found that, during a long process (>2 h), the initially formed *trans*-isomer converts into a more stable *cis*-form. Therefore, a two-hour reaction was optimal for obtaining *trans*-piperidinones with a diastereomeric ratio up to 3:1.

Yamazaki et al. carried out the intramolecular cyclization of alkene group-bearing amides by hydride transfer [77] (Figure 17). The formation of pipiridines with tret-amino groups proceeds efficiently in polar solvents such as DMSO, DMF, etc. However, the reaction is water-sensitive. The presence of water can lead to the loss of tertiary amino groups and the formation of a by-product with an alcohol residue.

Bower et al. showed the enantioselective intramolecular 6-exo aza-Heck cyclization of alkenylcarbamates [78] (Figure 18). The main feature of the reaction is redox neutral conditions. Therefore, it can be used for the synthesis of a wide range of substrates vulnerable to oxidation. The chiral P-O ligand provides a high selectivity of the reaction. Moreover, flexible conditions and the absence of an oxidizing agent allow for the use of a palladium catalyst for further one-pot cross-coupling.

Sadanandan and Gupta found an atypical intramolecular cyclization of β-lactams with an alkene residue [79] (Figure 19). The reaction pathway changes from 5-exocyclization to 6-endo cyclization to form piperidine rings, contrary to Baldwin’s rule. This is due to the restraint of the double bond and the dichloromethyl radical group by the lactam ring from the formation of a convenient intermediate for 5-exo cyclization in the transition state.

Shi et al. established a novel method of light-mediated intramolecular radical carbocyclization [80] (Figure 20). The process was developed to obtain fluorine derivatives of heterocyclic compounds from vinylidenecyclopropanes under the action of visible light radiation.

An interesting method of alkene cyclization through the S_N_2-reaction was presented by Kim et al. [81] (Figure 21). The chirality of the substrate was almost completely preserved. This effect is called memory of chirality (MOC). MOC was popularized by Fuji in the early 1990s and has been widely adopted ever since [82,83,84,85], including for obtaining heterocyclic compounds [86,87,88,89].

#### 2.2.2. Diene Cyclization

The Zhou group presented a highly enantioselective method for the intramolecular hydroalkenylation of 1,6-ene-dienes using a nickel catalyst and a chiral P-O ligand (Figure 22) [90]. The reaction provides a regioselective mild method for the preparation of six-membered *N-* and *O-*heterocycles with an aromatic substituent and an off-cycle double bond. According to the authors’ assumptions, the nickel catalyst is coordinated on diene and then incorporated into the double bond closer to the aromatic substituent, forming a more stable allylic intermediate, which undergoes cyclization.

Mori et al. proposed a method for the double C-H functionalization/cyclization of 1,3-ene-dienes with electron-withdrawing groups via a hydride shift (Figure 23) [91]. The process is initiated by chiral magnesium biphosphate, triggering two successive 1,5-H shifts to form two cycles. The stereoselectivity of the process can be increased by using an achiral ytterbium catalyst for the second cyclization. With this approach, the diastereoselectivity increases fourfold, while high yields are maintained. The method is suitable for obtaining substituted tricyclic systems.

The Yu group discovered the unexpected cycloisomerization of 1,7-ene-dienes (Figure 24) [92]. After a series of experiments, the authors concluded that the length of the substrate for this reaction plays a critical role. Thus, 1,6-diens give a [4 + 2] cycloaddition product under the described conditions. The approach is excellent for obtaining a wide range of *trans*-divinylpiperidines.

Feng et al. described a new intramolecular Alder-ene reaction of 1,7-dienes using a nickel catalyst (Figure 25) [93]. The reaction is a convenient way to obtain not only piperidines but also hydroquinoline, chromane, and thiochromane derivatives with high diastereo- and enantioselectivities. Magnesium and copper complexes are also suitable for the abovementioned method. However, nickel afforded much higher enantioselectivity values.

Mykhailiuk et al. proposed a photochemical method for obtaining bicyclic piperidinones from dienes via [2 + 2] intramolecular cycloaddition [94] (Figure 26). The resulting moieties can be easily converted into piperidines by reduction. Moreover, the reaction is scalable and proved to be useful for the synthesis of a key component analog of the antischizophrenia agent Belaperidone.

#### 2.2.3. Alkyne Cyclization

Saikia et al. proposed a carbenium ion-induced cyclization of alkynes by the action of ferric chloride, which played a dual role as Lewis acid as well as nucleophile (Figure 27) [95]. The method is suitable for obtaining *N-*heterocycles with alkylidene moieties. Of particular note, the reaction was E-selective for piperidines and Z-selective for pyrrolidines. The authors suggest that stereoselectivity depends on the attack of the chloride ion.

Takahashi et al. developed the radical cyclization of haloalkynes to produce five- and six-membered *N-*containing heterocycles [96] (Figure 28). A halogen at the triple bond affects the reactivity of the substrate and the stereoselectivity. Thus, an E/Z ratio of 5:1 was observed when using a phenyl substituent instead of a halogen.

You et al. achieved a gold-catalyzed intramolecular cyclization/dearomatization of β-naphthol derivatives with a terminal alkyne group [97] (Figure 29). An in situ-generated gold(I) complex activates the triple bond with further 6-exo-dig cyclization and protodemetalation to afford the spironaphthalenone product. It is interesting to note that the authors did not observe the expected competitive process—*O-*cyclization.

Kamimura et al. developed a new method for the synthesis of polysubstituted alkylidene piperidines from 1,6-enynes via intramolecular radical cyclization (Figure 30A) [98]. Triethylborane served as a radical initiator. The authors suggest the presence of a complex radical cascade, including two successive cyclizations (5-exo-dig and 3-exo-trig), cyclopropane cleavage to form a six-membered ring, and *cis*-selective hydrogen abstraction. Wang et al. used a similar approach to acquire iodo-homoallylic alcohols bearing piperidine rings (Figure 30B) [99]. The cyclization followed the 6-endo-trig pathway using acetonitrile as the solvent and the 5-exo-trig using methanol to afford the piperidine and azobicyclic (pyrrolidone/cyclopropane) derivatives, respectively. Therefore, the reaction is regioselective. However, all the resulting products are obtained as racemates.

Gharpure et al. described piperidine synthesis through the intramolecular 6-endo-dig reductive hydroamination/cyclization cascade of alkynes (Figure 31) [100]. The reaction proceeds via acid-mediated alkyne functionalization with enamine formation, which generates iminium ion. Subsequent reduction leads to piperidine formation. However, strong electron-releasing substituents (such as 4-OMe) at the aryl ring gave hydrolyzed derivatives instead of desired piperidines, while an electron withdrawing NO2 did not participate in the reaction at all.

Zeni et al. outline more information about the synthesis of six-membered *N-*heterocycles from alkynes in a big review [10].

#### 2.2.4. Radical-Mediated Amine Cyclization

Bruin et al. developed a new radical intramolecular cyclization of linear amino-aldehydes using a cobalt(II) catalyst [101] (Figure 32). The reaction proceeds in good yields and is effective for the production of various piperidines and pyrrolidones. However, during the synthesis of piperidines, the appearance of a by-product in the form of the corresponding linear alkene is observed. The authors suggest the presence of a competitive process between the radical rebound and 1,5-H-transfer, which results in the formation of a by-product.

Muñiz et al. developed two variants of the intramolecular radical CH-amination/cyclization of linear amines with electrophilic (aromatic) groups: anodic C-H bond activation through electrolysis (Figure 33A) [102] and both N-F and C-H bond activation through copper catalysis (Figure 33B) [103]. In the first variant, due to the single electron transfer, a radical cation is formed, which subsequently transforms into a benzyl radical after deprotonation. Further electron transfer results in a benzyl cation that reacts rapidly with tosylamide to give a heterocycle. The second method includes substrate coordination on a copper catalyst and further N-F bond cleavage via a single electron transfer with *N-*radical formation (N-F activation). Then, C-H activation occurs via fluorine-assisted hydrogen atom transfer, benzylic radical formation, etc. Wang et al. performed a similar approach [104] (Figure 33C). The developed method of radical cyclization makes it possible to obtain piperidines via 1,6-hydrogen atom transfer.

Another example of the *N-*radical approach to piperidines was represented by Nagib et al. (Figure 34) [105]. First, the enantioselective cyanidation of fluorosubstitued amines was carried out using a chiral copper(II) catalyst. The resulting aminonitriles underwent cyclization to chiral piperidines in the presence of DIBAL-H. The further optimization of the conditions made it possible to carry out the first asymmetric synthesis of an anticancer drug, Niraparib.

#### 2.2.5. Other Cyclizations

Line accomplished the enantioselective multistage synthesis of (3S, 4R)-3-hydroxypiperidine-4-carboxylic acid, including the key one-pot azide reductive cyclization of aldehyde (Figure 35) [106]. The resulting intermediate can be used for further modification and for obtaining various analogs of the final product.

Anderson et al. carried out the diastereoselective reductive cyclization of amino acetals prepared by the nitro-Mannich reaction (Figure 36) [107]. The diastereoselective Mannich reaction (first step) between functionalized acetals and imines is used to control the stereochemistry of piperidines, which is retained during reductive cyclization (second step).

Zu et al. developed a desymmetrization approach for piperidine synthesis through the selective lactam formation (Figure 37) [108]. The authors extended the scope of this method for the synthesis of the γ-secretase modulator. 

Song et al. developed a one-pot cyclization/reduction cascade of halogenated amides (Figure 38) [109]. Trifluoromethanesulfonic anhydride was used for amide substrate activation. Then, sodium tetrahydroborate was applied for imide ion reduction, followed by intramolecular nucleophilic substitution/cyclization. The reaction scope covers piperidines as well as pyrrolidines.

Darcel et al. developed the iron-catalyzed reductive amination of ϖ-amino fatty acids (Figure 39) [110]. Phenylsilane plays a key role in the reaction: it promotes the formation and reduction of imine, initiates cyclization, and reduces the piperidinone intermediate with iron complex as a catalyst. The method is efficient for the preparation of pyrrolidines, piperidines, and azepanes.

Morken et al. constructed piperidines by the intramolecular amination of methoxyamine-containing boronic esters (Figure 40) [111]. The reaction proceeds via N-B bond formation and the 1,2-metalate shift within the boron-intermediate. Precursors can be easily obtained through the Mitsunobu reaction. Moreover, this method was optimized for boronate-containing azacycles synthesis.

### 2.3. Intermolecular Cyclization (Annulation)

The intermolecular annulation process consists in the formation of a cycle of two or more components. Therefore, it can be divided into two-component reactions and multicomponent reactions, which will be discussed further.

#### 2.3.1. Two-Component Reactions

In the two-component intermolecular preparation of piperidines, the formation of two new bonds’ combination is mainly observed: C-N and C-C or two C-N. The condensation of amines with either aldehydes or ketones and with the further reduction of the imine group, also known as reductive amination, is one of the commonly used methods of C-N bond formation. This approach was applied mostly in [5 + 1] annulations.

A hydrogen borrowing the [5 + 1] annulation method was reported by Donohoe et al. (Figure 41) [112]. The mechanism includes two iridium(III)-catalyzed sequential cascades of hydroxyl oxidation, amination, and imine reduction by hydrogen transfer via a metal catalyst. The first amination is intermolecular (hydroxyamine intermediate formation), and the second is intramolecular. Thus, two new C-N bonds are formed. This approach enables the stereoselective synthesis of substituted piperidines. Moreover, the use of water as a solvent prevents the racemization of enantioenriched substrates, providing a route to highly enantioselective C4-substituted piperidines (Figure 41B).

Sather et al. proposed a new approach to piperidines through a combination of reductive amination and IMAMR (see Section 2.2.1) [113]. The reaction possesses predictable diastereoselectivity. Whenever ketone is used as a substrate, *trans*-selectivity is observed, and for aldehyde, the process is *cis*-selective, with a dr up to 20:1 and 1:12, respectively (Figure 42).

Griggs et al. further explored a stereoselective route to non-symmetrical spiropiperidinons via a similar condensation/intramolecular cyclization cascade (Figure 43) [114]. The C-H acid center acted as an active cyclization site. The enantioselective synthesis of piperidines via the 1,2-diamination of aldehydes was reported by Ramapanicker et al. (Figure 44) [115]. The initial alpha-amination of aldehyde with dibenzyl azodicarboxylate (DBAD) was followed by reductive amination/cyclization. This method is designed for the stereoselective synthesis of amine-substituted piperidines.

Double reductive aminations are an effective route to piperidines. Thus, Jiang et al. proposed the highly selective ruthenium(II)-catalyzed double reductive amination/hydrosylilation of glutaric dialdehyde and aniline derivatives (Figure 45A) [116]. It is worth noting that the method is suitable only for amine substrates with the p-π conjugation effect. Rao et al. achieved a similar process through a microwave-mediated Leuckart reaction (Figure 45B) [117]. Piperidines were afforded from diketone and aryl ammonium formate, which played a double role as a nitrogen source and reductant. Kiss et al. developed the stereocontrolled synthesis of fluorine-containing piperidines from racemic cyclic diols (Figure 45C) [118]. The chain of reactions proceeded through an oxidative ring opening to form an unstable dialdehyde, which immediately underwent double reductive amination/ring closure with a fluorine-containing quaternary ammonium salt. Sodium cyanoborohydride was used as a reducing agent.

The aza-Prins reaction is another efficient way to accomplish piperidine synthesis [119]. Thus, Li et al. proposed the aza-Prins cyclization of homoallylic amines with aldehydes promoted by the NHC-Cu(I) complex and ZrCl_4_ (Figure 46A) [120]. After the aldehyde group activation by zirconium chloride, an iminium intermediate was obtained; it further underwent 6-endo-trig cyclization with the formation of a carbocation in the 4-position. The *trans*-selectivity of the reaction is explained by steric hindrance. Thus, a nucleophilic attack of the chloride ion from the axial side is less favored. Rajasekhar et al. carried out a similar route with epoxides (Figure 46B) [121]. There, niobium pentachloride served as Lewis acid and a chlorine source.

Martin et al. developed [5 + 1] aza-Sakurai cyclization for spiropiperidines (Figure 47A) [122] and piperidines (Figure 47B) [123] synthesis using amines with cyclic ketones and aldehydes, respectively. The process includes an intermolecular reaction of imine formation through condensation, followed by intramolecular cyclization. The resulting piperidines carry three functional centers (NH, olefin, aromatic groups) suitable for further derivatization.

Xu et al. displayed a [5 + 1] acid-mediated annulation through the aza-Pummerer approach (Figure 48) [124]. The acid complex used promotes the formation of the carbenium ion from dimethyl sulfoxide via Pummerer fragmentation. Moreover, hydrochloric acid acts as a chlorine source. Thus, three new bonds were formed: C-N, C-C and C-Cl.

Kim et al. proposed the metal-free *N-*heterocyclization of arylamines with cyclic ethers (Figure 49) [125]. The reaction took place with the use of phosphoryl chloride, which initiated the ether ring-opening and the subsequent piperidine ring-closure, with two new C-N bond formations overall.

In addition to [5 + 1] annulation, [4 + 2] and [3 + 3] reactions have also been applied in piperidine synthesis.

Wu et al. represented another variant of radical-mediated cyclization (Figure 50A) [126]. As in the abovementioned works (see chapter 2.2.4), the copper catalyst initiates *N-*radical formation, and afterwards, the 1,5-HAT carbon radical is captured by CO and Cu(II) species to form the proposed intermediate, followed by an intramolecular ligand exchange and reductive elimination/cyclization. It is worth noting that the intermediate could undergo an intermolecular exchange with alcohols giving esters. Atobe et al. introduced a flow microreactor for the radical electroreductive cyclization of imines with terminal dihaloalkanes (Figure 50B) [127]. The reaction goes through radical anion formation, which is involved in the intermolecular nucleophilic attack on the terminal dihaloalkane. Then, the earlier-formed *N-*radical undergoes one-electron reduction and intramolecular cyclization.

Balakumar et al. presented a methodology for the synthesis of 2-substituted carboxypiperidines from amino acids with a known stereochemistry (Figure 51) [128]. The cycle was formed in the alkylation stage of the amine with dihaloalkane in high yields.

Gulías et al. developed a palladium that promoted the formal [4 + 2] oxidative annulation of alkyl amides and dienes (Figure 52A) [129]. A feature of this reaction is the activation of C(sp3)-H bond, which is atypical for cycloadditions. The authors proposed that the activation and cleavage of C(sp3)-H bond were processed through migratory insertion/reductive elimination mechanisms. The diene substrate is essential to directing the process towards reductive elimination. Guo et al. showed the organocatalytic [4 + 2] annulation of *N-*tethered enones and dicyanoalkenes (Figure 52B) [130]. The reaction tolerated a broad scope of substrates and possessed high yields and great diastereoselectivity.

The [3 + 3] cycloaddition method has increasingly attracted the attention of the scientific community for the synthesis of heterocyclic compounds in recent years [131,132]. Thus, Yang et al. described the regioselective [3 + 3] annulation of enones with α-substituted cinnamic acids (Figure 53) [133]. The reaction proceeded through intermolecular Michael addition, decarboxylation, and intramolecular lactamization/cyclization.

#### 2.3.2. Multicomponent Reactions

By definition, multicomponent reactions (MCRs) include one-pot processes in which three or more components interact to form a target compound containing, in its structure, the majority of atoms of all starting substances (Figure 54) [134,135]. Reactants are mixed in one reaction vessel, without the introduction of additional reagents during the reaction process. MCRs have a number of significant advantages compared to two-component reactions: the simplicity and availability of reagents, the reduction in the number of synthesis stages, the simplification of the process of isolating final compounds, the reduction in solvent consumption, and, as a result, their environmental friendliness and higher efficiency.

Multicomponent synthesis is firmly rooted in organic chemistry as the main way to obtain various classes of compounds. MCRs are often used in the complete synthesis of complex natural compounds, require a minimum set of starting materials, and make it possible to obtain extensive libraries of compounds that have a structure similar to that of biologically active drug components.

Reactions of this type make a great contribution to the convergent synthesis of complex organic molecules, which are of great importance for the pharmaceutical industry, biochemistry, and research in the field of medicine [136].

MCRs include a variety of methods such as the Strecker synthesis, the Hanch synthesis of dihydropyridins and pyrroles, the Biginelli reaction, the Mannich reaction, the Ugi reaction, the preparation of imidazoles according to Radziszewski, the Passerini reaction, and many others [137,138,139,140,141]. MCRs have an indisputable importance in modern organic chemistry.

One of the first examples of the multicomponent synthesis of compounds containing a piperidine fragment is shown in the work of Guareschi in 1897 (Figure 55) [142]. The synthesis was carried out by an MCR between butanone, ethyl cyanoacetate, and an alcohol solution of ammonia to obtain a cyclic imide. The yield of the final imide was not reported in the work.

This work was followed by a series of publications devoted to the preparation of Guaresi imides in a multicomponent variant using an alcoholic solution of ammonia, ethyl cyanoacetate, aldehydes and ketones of various structures [143,144].

In recent years, the number of publications devoted to the multicomponent synthesis of piperidine-containing compounds is relatively small in comparison with multistage methods. However, the scientific world community still offers new approaches to solving the problems of multistage syntheses using MCRs. Further, we highlighted some of the last year’s works related to the multicomponent synthesis of various substituted piperidine cycles. It is worth noting that MCRs are more often used for obtaining piperidine cycles with double bonds (tetrahydropyridines). However, this is beyond the scope of this review. You can find more examples of this method in recently cited publications [3,145,146,147,148,149,150,151,152,153,154].

Islam et al. invented a novel polystyrene ferric-based azo-catalyst for the highly effective synthesis of spiropiperidine derivatives (Figure 56) [155]. Primary aromatic amine was used as a nitrogen source (Figure 3). The described catalyst worked much better than raw FeCl_3_*6H_2_O, allowing for high yields, the full conversion of reactants without any heating, and the recyclability of the catalyst.

In the work of Ahmad et al., the obtainment of the same compounds was achieved with a chitosan-supported ytterbium heterogeneous nano-catalyst (Figure 57) [156]. The Yb/chitosan catalyst was active with acyclic methylene compounds and dimedons in the same conditions. Thus, the method used is also suitable for the preparation of piperidines without spirocyclic substituents.

In the described methods, piperidine cycle formation was reached through a domino process of different mechanisms including Knoevenagel condensation, Michael addition, and two consecutive Mannich reactions. Such cascade is common for most of the multicomponent synthesis of piperidines (Figure 58). More information about MCRs combinations of «name reactions» can be found here [157].

For example, Vereshchagin’ group developed a stereoselective variant of the abovementioned approach for the synthesis of poly-substituted piperidines using ammonium acetate both as a nitrogen origin and catalyst for CH-acid deprotonation [158,159,160,161,162,163]. Ammonium acetate plays a key role in the synthesis of various compounds in organic chemistry, including its use in MCRs. Reactions with ammonium acetate are widely adopted [164]. Therefore, the preparation of γ-lactams [165], furo [3,2-c]chromen-4-ones [166], imidazoles [167,168,169,170,171], triarylpyridines [172], substituted 3-cyanopyridines [173], dihydropyridines [174,175] of various structures, and many other compounds through multicomponent processes is carried out using ammonium acetate. The ease of its use and its commercial availability make ammonium acetate one of the most common reagents utilized to introduce one or more nitrogen atoms into the structure, which is essential in multicomponent processes [176].

First, the authors obtained a series of piperidine diastereomers with aromatic substituents from benzaldehydes, malononitrile and ammonium acetate in pseudo six-component synthesis (Figure 59A) [158]. The reaction showed high yields and great stereoselectivity. Then, the authors added formaldehyde to the MCR system in order to achieve a product without the third aromatic ring (Figure 59B) [159]. The researchers proposed that formaldehyde undergoes Knoevenagel condensation instead of benzaldehyde, which, in the original method, participates only in the Mannich reaction.

The same approach was applied when the product of Knoevenagel condensation was obtained separately (Figure 60) [160,161]. In this case, the reaction goes through a Michael/Mannich cascade with four new bonds formed. However, this approach has several limitations. When trying to obtain piperidines with different substituents in the benzene fragments, a mixture of a two-to-one ratio is observed using ammonium acetate, and one of a three-to-one ratio is observed using aqueous ammonia (Figure 60C) [161]. The authors suggest that the by-product is formed due to competitive mechanisms. In parallel with the Michael/Mannich cascade, retro-Knoevenagel condensation occurs with the formation of benzaldehyde, which then participates in the further transformation.

The scope of the described reaction includes the synthesis of piperidinols (Figure 61) [162,163]. The use of esters of 3-oxocarboxylic acid instead of aldehyde resulted in a product containing four stereocenters. Moreover, the resulting piperidinol undergoes dehydration in an acidic environment to form 1,4,5,6-tetrahydropyridine. The process passes through a previously unknown isomer of 3,4,5,6-tetrahydropyridine. You can read more about the mechanism and kinetics of the reaction here [162].

Rashinkar and colleagues developed a new green approach to the synthesis of piperidinols [177]. The authors used previously discovered unexpected cyclization in an amine exchange reaction between primary amines and Mannich bases. Thus, they obtained a broad range of substituted piperidinols by means of water-mediated intramolecular cyclization that occurs after *bis*-aza Michael addition (Figure 62). The products were afforded in slightly better yields when the Mannich base contained electron-withdrawing substituents. In addition, the resulting compounds showed modest anthelmintic activity.

An interesting approach to piperidines was proposed by Tehrani et al. (Figure 63) [178]. The authors used calcium carbide as an acetylene source and drying agent promoting the formation of cyclic ketimine from chlorinated ketone and amine. The researchers also proposed that the formed acetylene could react with a catalyst—cuprum(I) iodide. The addition of the resulting cuprum carbide to ketimine and protonation leads to product formation and catalyst regeneration. Piperidine with terminal alkyne can further be used in different cross-coupling reactions.

Vereshchagin et al. also developed a series of poly-substituted piperidinons by means of pseudo four-component synthesis (Figure 64) [179,180]. All of the studied MCRs lead to one diastereomer with modest to high yields. Piperidinons, like piperidines, can be synthesized by the Michael/Mannich cascade (Figure 64A) and Knoevenagel/Michael/Mannich cascade (Figure 64B). Thus, the used approach proved to be useful for a broad variety of piperidine scaffolds.

Savithiri et al. used a similar MCR with ammonium acetate to synthesize naphthyl-substituted piperidons (Figure 65A) [181]. The picrates derived from the obtained piperidons possessed relatively good antibacterial and antiviral properties. Ilangeswaran and colleagues designed a greener MCR for acquiring piperidinons using a glucose-urea deep eutectic solvent (Figure 65B) [182].

Lin and Yan et al. represented a view on the solvent-free piperidine-promoted synthesis of piperidons from 2-cyanoacetamides and ketones (Figure 66) [183]. The scope of the reaction covers piperidons (Figure 66A) as well as spiropiperidinones (Figure 66B) derivatives. It is worth noting that the reaction leads to 2-oxo-1,2,3,4-tetrahydropyridine if the amide group in 2-cyanoacetamides possesses either an alkyl or hydrogen substituent instead of aryl. Therefore, the described MCR is regioselective.

## 3. Pharmacological Applications of Piperidine Derivatives

The piperidine cycle is utterly common in pharmaceuticals. Its derivatives are used in over twenty drug classes [184], including anticancer agents [18,185,186,187,188,189], drugs for Alzheimer’s disease therapy [46], antibiotics [190], analgesics [17,191], antipsychotics [19,192,193], antioxidants [15,194], etc. (Figure 3).

Moreover, piperidines are also a part of many alkaloids showing biological activity (Figure 4). For example, the well-known atropine (used clinically for the treatment of vomiting, nausea, and bradycardia [195]; an effective agent for slowing the development of myopia [196]) and morphine (analgesic for severe pain relief [197]; used as a third-line therapy in the treatment of neuropathic pain [198]) contain a fused piperidine ring.

Piperine, a derivative of piperidine and the main active chemical component of black pepper, is attracting more and more attention from researchers, despite the fact that it was discovered more than 200 years ago. It is believed that piperine has a broad scope of beneficial biological properties, from antibacterial to anticancer [199,200,201,202,203]. Aloperine and Matrine—an alkaloid of the Sophora containing two fused piperidine rings at once—and their derivatives showed antiviral, anti-inflammatory, and antitumor properties [204,205]. Febrifugine and its synthetic analog halofuginone are efficiently used as antiparasitic drugs [20].

Along with already known drugs, the scientific community constantly proposes new biologically active piperidine scaffolds. Further, we will discuss recent discoveries in the biological evaluation of synthetic potential drugs containing the piperidine moiety. Particular attention was paid to four pharmaceutical groups: cancer (pro-tumorigenic receptor inhibitors, apoptosis initiators), infectious and parasitic diseases (biocides), Alzheimer’s disease (anticholinergics), and neuropathic pain (analgesics). The choice of drug groups was based on current trends and relevance in the medical community.

### 3.1. Cancer Therapy

Cancer is one of the biggest health problems worldwide, with nearly 10 million deaths reported in 2020 according to WHO. A lot of resources are spent on the development of new drugs for fighting cancer, but despite all efforts, innate and acquired resistance mechanisms are often observed [206]. Therefore, screening for new developments and breakthroughs in this area is very important and relevant.

Piperidine moieties are often used in anticancer drug construction [185,189]. Herein, the recent proposals and developments of scientists on this subject will be briefly discussed.

Arumugam et al. synthesized spirooxindolopyrrolidine-embedded piperidinone **1** with potential anticancer activity through three-component 1,3-dipolar cycloaddition and subsequent enamine reaction [207]. The resulting product showed slightly better cytotoxicity and apoptosis induction in the FaDu hypopharyngeal tumor cells model than the reference drug bleomycin. The authors followed the “escape from flatland” approach, which was popularized throughout recent years [208,209,210] and was successfully used in the development of anti-cancer agents [211,212,213]. This approach suggests that more saturated and three-dimensional structures will interact better with binding sites of proteins. Therefore, the authors reasoned that the spirocyclic structure played a key role in the biological activity of compound **1**.

Li et al. developed IκB kinase (IKKb) inhibitor **2** as an EF24 analog [214]. EF24 is a piperidinone derivative with potential activity against lung, breast, ovarian, and cervical cancer [215,216]. The activation of IKKb is one of the major factors of NF-κB transcription, which induces chronic inflammation in carcinomas, leading to desmoplasia and neoplastic progression [217]. The new analog **2** possessed better IKKb inhibitory properties than the reference drug. The active component with the piperidine moiety developed a stable hydrophobic interaction with the IKKb catalytic pocket. The structure–activity relationship shows that the presence of a nitrogen atom in the cycle is optimal, and any substitution in the ketone bridge is not favorable for IKKb inhibition.

A series of 2-amino-4-(1-piperidine) pyridine derivatives **3,** as the clinically Crizotinib-resistant anaplastic lymphoma kinase (ALK) and c-ros oncogene 1 kinase (ROS1) dual inhibitor, was designed by Zhang et al. [218]. ALK was originally discovered in anaplastic large cell lymphoma as a transmembrane receptor tyrosine kinase [219]. It was found that ALK is involved in the development of non-small cell lung cancer, neuroblastoma, diffuse large B-cell lymphoma, anaplastic thyroid cancer, rhabdomyosarcoma, ovarian cancer, esophageal squamous cell, colorectal, and breast carcinomas, etc. [220,221]. ROS1 was discovered more recently as a similar enzyme to ALK. ROS1 rearrangements were identified in glioblastoma, cholangiocarcinoma, gastric cancer, ovarian cancer, soft-tissue sarcomas, breast cancer etc. [222]. Crizotinib—the first approved ALK/ROS1 dual inhibitor—also includes the piperidine moiety [223]. Despite the fact that the piperidine fragment did not form any bonds with the active site of the receptor, its introduction was the most optimal for the desired pharmacological properties compared to other substituents. In the case of substance 3, piperidine derivatives (namely, (S or R)-ethyl piperidine-3-carboxylate) were used as the starting components for chiral optimization [218]. The piperidine ring was essential for chiral optimization. Piwnica-Worms et al. obtained radiolabeled fluoro-analogs of the commercial ALK inhibitors crizotinib **4**, alectinib **5**, and ceritinib **6** [224]. The products have a potential for brain metastases treatment due to their enhanced CNS pharmacokinetic properties. It is worth noting that the introduction of fluoroethyl groups did not affect the inhibitory properties of parent drugs, while it enhanced their ability to pass through blood–brain barrier.

Onnis et al. conducted a synthesis of benzenesulfonamide with a piperidinyl-hydrazidoureido linker as potent carbonic anhydrase (CA) II (**7**), IX (**8**), and XII (**9**) inhibitors [225]. CAs are metalloenzymes localized in the cytosol, mitochondria, membranes, and secreted substances of living organisms. CAs are involved in the catalysis of chemical processes (the hydration of carbon dioxide to bicarbonate, the conversion of cyanate to carbamic acid, etc.) and esterase activity [226]. Two out of the sixteen known types of CA (CA IX and CA XII) are found in vertebrate tumor cells. Their inhibition is an effective way to control the growth, progression, and metastasis of cancerous tumors [227]. Currently, the leading compound among CA IX and XII inhibitors is SLC-0111, which is in phase I/II of clinical trials for the management of hypoxic tumors [228,229]. The authors used SLC-0111 as the parent drug, incorporating a piperidinyl-hydrazidoureido linker in its structure to improve binding selectivity with CA. The piperidine residue was also introduced to assess rigidity [225].

Benzoylpiperidine scaffold **10** with antitumor activity via monoacylglycerol lipase (MAGL) inhibition was constructed by Granchi et al. [230,231]. Fluorine atoms and the meta-substitution of the benzene ring enhanced the inhibition properties. MAGL is responsible for the inactivation of the brain’s endocannabinoid 2-arachidonoylglycerol. Moreover, MAGL indirectly controls the levels of free fatty acids, as well as other lipids with pro-inflammatory or pro-oncogenic effects, therefore causing pain and cancer progression [232]. Sekhar et al. developed spirochromanone analog **11** with significant activity against the breast cancer cell line and Murine melanoma, as well as the ability to induce apoptosis [233]. The authors combined known pharmacophore structures to achieve the best anti-proliferative and anti-cancer effects.

Jeong et al. synthesized piperidine-embedded anticancer agents with particularly good activity on androgen-refractory cancer cell lines (ARPC) [234]. The authors showed that compound **12** was a ligand to the M3 muscarinic acetylcholine receptor (M3R), which is presented in ARPC (Figure 5). M3R activation stimulates cell proliferation, resistance to apoptosis, and metastasis and is responsible for the early progression and invasion of colorectal cancer tumors [235,236,237].

### 3.2. Alzheimer Disease Therapy

Alzheimer disease is one of the most lethal and burdening illnesses of the last century. It has no definite treatment other than symptomatic treatment and results in death 6 years after diagnosis, on average [238]. The oldest theory of Alzheimer’s disease is the cholinergic hypothesis, which suggests that the illness is caused by the loss of cholinergic innervation [239].

The neurotransmitter acetylcholine is one of many vital components for normal brain function. Deficiency of the cholinergic system has been observed in the brains of Alzheimer’s disease patients, leading to the pathophysiology of learning and memory impairment [240]. The main goal of modern therapy is to maintain the level of acetylcholine through the inhibition of cholinesterases: acetylcholinesterase (ACHe) and butyrylcholinesterase (BuCHe) [241]. Currently, the leading drug among acetylcholinesterase inhibitors is Donepezil, a piperidine derivative.

Martinez et al. proposed indolylpiperidine analog **13** of Donepezil [242]. The active agent was capable of inhibiting both acetylcholinesterase (AChE) and butyrylcholinesterase (BuChE) enzymes. Moreover, the authors discovered unusual conformational changes in the molecule depending on the binding site. Thus, compound **13** was extended in AChE interaction and flopped in BuChE interaction. Liu et al. expanded this field with 4-N-phenylaminoquinoline derivative 14 [243] via piperidine moiety introduction to a previously reported lead compound [244]. Piperidine incorporation improved the brain exposure of the resulting dual inhibitor. In addition, the compound showed antioxidant and metal chelating properties.

In 2018, Gobec et al. designed selective BuChe inhibitor **15** [245]. Two cationic nitrogen atoms were essential for selectivity and good inhibition properties. Further, the authors conducted a detailed structure–activity relationship study of *N-*alkylpiperidine carbamates [246]. Structures with an *N-*benzyl moiety were superior in cholinesterase inhibition, and a terminal alkyne group was essential for efficient monoamine oxidase B inhibition. Thus, compounds **16–18** were selected as the best in the series. Moreover, a study by Malawska and Gobec outlined a multi-targeted approach to Alzheimer’s disease treatment. Novel 1-Benzylpyrrolidine-3-amine derivatives with piperidine groups **19** and **20** expressed both antiaggregatory and antioxidant effects [247]. Along with dual cholinesterase inhibition, compounds **19–20** also targeted the beta secretase enzyme.

Beta secretase is also known as beta-site amyloid precursor protein cleaving enzyme-1 (BACE-1). It has been established that the inhibition of BACE-1 prevents the accumulation of amyloid beta [248,249]. According to the current concept of Alzheimer’s disease based on the amyloid hypothesis, deposits of amyloid beta and tau proteins cause neurodegeneration and cognitive impairment [250].

The benzyl-piperidine group (Donepezil-like) is often a necessary part for the successful inhibition of cholinesterase receptors. The AChE enzyme includes two active anionic binding sites: catalytic and peripheral. The benzyl-piperidine group provides good binding to the catalytic site, interacting with Trp84, Trp279, Phe330, and Phe331 [251]. Therefore, the selection of various substituents on top of the benzyl-piperidine residue is a well-established approach in the synthesis of new active agents for combatting Alzheimer’s disease. Thus, pyrrolizine 21 [252], fluorine 22 [253], thiazole 23 [254], indoline 24 [255], benzofuran 25 [256], thiophene 26 [257], and chromene 27 [258] groups have been effectively incorporated and biologically evaluated by various authors. The structure–activity relationship is shown in Figure 6.

It is worth noting that research on multifunctional active agents is prevalent compared to compounds that affect only one target. Therefore, along with inhibitors of cholinesterase receptors, attention is also paid to inhibitors of monoamine oxidase **16–17**, **26** [246,257], amyloid beta and tau protein aggregation **19–20**, **22**, **25–26** [247,253,256,257], BACE-1 **19–20**, **22**, **24**, **27** [247,253,255,258], as well as the presence of anti-inflammatory 26 [257], anti-radical **19–20** [247], and antioxidant properties **19–20**, **22**, **26** [247,253,257]. 

Drawing conclusions from the analyzed data, it can be said that the piperidine group affects the inhibition of cholinesterase receptors and serves as a constructing (linker) part.

### 3.3. Biocides

Biocides are chemical compounds designed to neutralize, suppress, or prevent the action of harmful organisms, namely, pathogenic bacteria, fungi, viruses, parasites, etc. [259]. As noted earlier, piperidine derivatives find use in this class of pharmaceuticals. 

In recent years, a number of works on the topic can be noted. However, due to the wide variety of human pathogens, it is not possible to point out one template structure for all types of activities.

Thus, piperidine moieties were represented in structures with antifungal properties **28–30**. Compounds containing tartaric acid fragment **28–29** inhibited chitin synthase, therefore suppressing a growth of five fungi strains (*C. albicans* ATCC 76615, *A. fumigatus* GIMCC 3.19, *C. albicans* ATCC 90023, *C. neofonmans* ATCC 32719, *A. flavus* ATCC 16870) [260]. The resulting compounds combined two pharmacophores: 2,8-Diazaspiro [4.5]decane-1-one and a tartaric acid residue with a substituted aminobenzene. When designing the structure, the authors were guided by the “escape from flatness” theory (which was mentioned earlier) and the enzyme inhibition potential of the chosen moieties. Long-tailed 4-aminopiperidines **30** have proven to be effective against fungi of the genus Aspergillus and Candida via fungal ergosterol biosynthesis inhibition [261]. Ergosterol is one of the most abundant fungal cell membrane sterols. It is responsible for membrane permeability and fluidity [262].

Piperidine-containing fluoroquinolones analogs, namely, bafloxacins **31**, were proposed by Liu et al. [263]. Along with the good inhibition values on MRSA, *P. aeruginosa*, and *E. coli*, the new compounds showed good biocompatibility and potential two-targeted action via cell walls destruction and interaction with IV-DNA and DNA gyrase. Piperidinyl “tails” structures **32** possessed inhibition properties against streptomycin-starved Mycobacterium tuberculosis 18b (SS18b) and H37Rv strains [264]. Compound **32** consists of various known tubercular pharmacofores with piperidine as a linker.

The benzyl-piperidines activity against different viruses was shown. Thus, 4,4-disubstituted *N-*benzyl piperidines **33** inhibited the H1N1 influenza virus through specific hemagglutinin fusion peptide interaction [265]. Nayagam et al. discovered the potential inhibitor of SARS-CoV2 with piperidine core **34** [266]. Compound **34** possessed a better binding affinity with the SARS-CoV2 main protease than Remdesivir, with five binding pockets interaction compared to two.

Compounds with a piperidine backbone structure have shown antiparasitic properties on *T. brucei* (the main cause of African trypanosomiasis) **35** [267] and *P. falciparum* (the cause of the deadliest type of malaria) **36–37** [268,269] (Figure 7).

### 3.4. Neuropathic Pain Therapy

Neuropathic pain occurs as a result of the pathological excitation of neurons in the peripheral or central nervous system, which is caused by neurological diseases with damage to peripheral fibers and central neurons [270]. The modern approach to the treatment of neuropathic pain includes three lines of pharmacotherapy. Most of the piperidine derivatives are part of opioids, which are the second and, in some cases, the third line of treatment [198].

Opioid receptors are divided into four similar types: μ-opioid (MOR), δ-opioid (DOR), κ-opioid (KOR), and nociceptin/orphanin FQ peptide receptor (NOP) [271,272]. MOR and DOR are the main targets of opioid agonists. MOR agonists cause euphoria and help with coping with stress; however, their use causes serious side effects and physical dependence, leading to overdose [273]. One of the main synthetic piperidine-containing opioids is fentanyl.

Most often, opioid derivatives serve as the starting point for the discovery of new types of analgesics. Thus, derivatives of norsufentanil with amino acids **38** were developed [274], the synthesis of the main metabolites of carfentanil **39** was reproduced [275], and new analogs of tramadol **40** were proposed [276]. All compounds showed a strong affinity for MOR.

In order to achieve a dual effect ligand, Lee et al. created a hybrid based on Pethidine, also known as meperidine, and a transient receptor potential cation channel subfamily V member 1 (TrpV1) antagonist **41** [277]. TrpV1 functions are widely linked to the generation of pain [278]. This combination potentially increases the anti-inflammatory effect and treatment efficiency.

Navarrete-Vázquez et al. developed a haloperidol analog **42** as a σ1 receptor antagonist [279]. The σ1 receptor plays a role in various regulatory processes, including pain reduction [280]. Therefore, Chen et al. proposed novel piperidine propionamide derivatives **43–44** as dual agonists of μ-opioid and σ1 receptors [281,282].

Lastly, CA inhibition is another prominent therapeutic target in neuropathic pain treatment. Thus, Supuran et al. synthetized piperidine-embedded 4-oxo-spirochromanes **45** with high activity against CA II and CA VII (Figure 8) [283].

## 4. Conclusions

Piperidines play an important role in both chemistry and medicine. In this review, attention has been paid to both of these applications of piperidines.

We can safely say that approaches to the piperidine synthesis have improved significantly in recent years. There is a shift towards shortening the synthesis steps number. Thus, methodologies are used for the simultaneous functionalization/hydrogenation of pyridines, the functionalization/cyclization of unsaturated amines, as well as multicomponent cascade reactions with the formation of several new C-N and C-C bonds at once. Scientists are successfully applying new routes for the stereoselective synthesis of piperidines, using catalysis with transition metal cores such as palladium, rhodium, ruthenium, nickel, chromium, platinum, cobalt, etc. complexes, as well as organocatalysts (amino-hydroquinine, chiral NHC-catalyst, boron complexes). Moreover, new green reactions have also been proposed: non-toxic iron catalysis, water-initiated processes, environmentally benign electrolytic methods, solvent-free reactions, etc.

In the medical part, examples of the pharmacological use of synthetic piperidines and natural piperidine derivatives in the composition of alkaloids are given. We have shed light on some of the latest scientific achievements in the synthesis of biologically active agents based on piperidines against socially significant diseases. Overall, piperidines can act both as a part of the pharmacophore, interacting directly with the active site of enzymes, and as a convenient building block to achieve the desired conformation and physical properties of pharmaceuticals.

The current review was designed to highlight the wide range of synthesis and medical applications of piperidines, as one of the most important groups of chemicals.

## Data Availability

Not applicable.

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
