# Peer review of "Piperidine Derivatives: Recent Advances in Synthesis and Pharmacological Applications"

_ijms, 2023, doi:10.3390/ijms24032937_

Round 1

Reviewer 1 Report

The authors have summarized the chemical methods of functionalization, synthesis of piperidine derivatives and pharmaceutical application of piperidines. Piperidines are one of important fragments in designing drugs. It might be very useful for designing synthetic scheme of piperidine containing compounds.  I would like to recommend this manuscript publish in present form.

Author Response

We thank the reviewer for the comments!

Reviewer 2 Report

Although I am a drug designer rather that a wet-lab medicinal chemist, I enjoyed reading this review - in particular its first (and most developed) organic synthesis part, which allowed me to catch up with the latest synthesis methods of piperidines (as I said, being only tangentially involved in actual medicinal chemistry, I do not follow organic synthesis literature regularly). However, the manuscript should be proofread by a native English speaker, to fix some of the occasionally weird expressions and replace colloquial terms such as "humonguous", which might not go well with readers who lost relatives to cancer.

I did not enjoy so much the second part, dedicated to biological activity "of the piperidines" - which is debatable, because, well - ligands "stand" against their biological target as a Whole, and it is never clear whether containing a piperidine was really needed to achieve the required binding properties or in vitro activity - or whether those compounds just contain piperidine because it's a popular moiety with synthetic chemists! You will agree with me that being often seen in drugs is not an automatic indicator of importance - after all, 99.99% of a drugs contain carbon atoms, but it is unlikely to conceive a review article entitled "The antibiotic effect of carbon atoms".  It would be thus nice to discuss, whenever possible, WHY that piperidine is important - whenever it is, as in some cases it may well not be (in the sense that many other organic fragments could have taken its place without compromising bioactivity). This information is of course not easy to find - one must dig into the analogue series leading to the drug, in order to see whether piperidine is (a) part of the pharmacophore, (b) a welcome trick to reduce ligand floppiness by cyclization (binding entropy loss minimization), (c) introduced in order to improve ADME/Tox properties of the drug, or (d) simply used because that's what the chemist could buy/liked to synthesize. Are there reasons for which piperidines  are better than other tertiary amines?

Author Response

We thank the reviewer for the comments!

Although I am a drug designer rather that a wet-lab medicinal chemist, I enjoyed reading this review - in particular its first (and most developed) organic synthesis part, which allowed me to catch up with the latest synthesis methods of piperidines (as I said, being only tangentially involved in actual medicinal chemistry, I do not follow organic synthesis literature regularly). However, the manuscript should be proofread by a native English speaker, to fix some of the occasionally weird expressions and replace colloquial terms such as "humonguous", which might not go well with readers who lost relatives to cancer.

Response: The review has been corrected by a native English speaker. Grammar corrections and re-phrased fragments are highlighted in yellow.

I did not enjoy so much the second part, dedicated to biological activity "of the piperidines" - which is debatable, because, well - ligands "stand" against their biological target as a Whole, and it is never clear whether containing a piperidine was really needed to achieve the required binding properties or in vitro activity - or whether those compounds just contain piperidine because it's a popular moiety with synthetic chemists! You will agree with me that being often seen in drugs is not an automatic indicator of importance - after all, 99.99% of a drugs contain carbon atoms, but it is unlikely to conceive a review article entitled "The antibiotic effect of carbon atoms".  It would be thus nice to discuss, whenever possible, WHY that piperidine is important - whenever it is, as in some cases it may well not be (in the sense that many other organic fragments could have taken its place without compromising bioactivity). This information is of course not easy to find - one must dig into the analogue series leading to the drug, in order to see whether piperidine is (a) part of the pharmacophore, (b) a welcome trick to reduce ligand floppiness by cyclization (binding entropy loss minimization), (c) introduced in order to improve ADME/Tox properties of the drug, or (d) simply used because that's what the chemist could buy/liked to synthesize. Are there reasons for which piperidines  are better than other tertiary amines?

Response: Thank you for the question! We believe, that piperidines can act both as a part of the pharmacophore, interacting directly with the active site of enzymes (for example, the interaction of the benzyl-piperidine residue with the catalytic binding site of acetylcholinesterase), and as a convenient building block to achieve the desired conformation and physical properties of pharmaceuticals.

In the revised version of the review, we have added more information about structure-activity relationships (mainly on figures 5-8), and tried to answer the question of what role piperidine plays in the described biologically active compounds.

We thank the reviewer for the comments!

Reviewer 3 Report

Dear,

The authors left much to be desired in some points, for example: the authors emphasized many synthesis mechanisms that are easily found in textbooks in the area and forgot to focus on the pharmacological application (they mentioned at the end only cancer therapy and Alzheimer disease therapy).

In fact, it has a lot of synthesis mechanisms and little written material on pharmacological activity (and poorly written in some parts). He doesn't just talk about cancer and alzheimer's, but also about biocidal activity (antifungal, antiviral, antiparasitic, antibacterial) and something about treating neuropathic pain. The problem is that he writes very little of it.
The authors cite several pharmacological applications of the derivatives, but discuss little about them. I think it would be interesting for him to focus on the MAIN pharmacological applications and go deeper into the discussion.  So the article will not be shallow in that part.

As I understand little about organic synthesis, I cannot say whether this part is good in the article. I just thought it was too long.

Author Response

We thank the reviewer for the comments!

The authors left much to be desired in some points, for example: the authors emphasized many synthesis mechanisms that are easily found in textbooks in the area and forgot to focus on the pharmacological application (they mentioned at the end only cancer therapy and Alzheimer disease therapy).

In fact, it has a lot of synthesis mechanisms and little written material on pharmacological activity (and poorly written in some parts). He doesn't just talk about cancer and alzheimer's, but also about biocidal activity (antifungal, antiviral, antiparasitic, antibacterial) and something about treating neuropathic pain. The problem is that he writes very little of it.
The authors cite several pharmacological applications of the derivatives, but discuss little about them. I think it would be interesting for him to focus on the MAIN pharmacological applications and go deeper into the discussion.  So the article will not be shallow in that part.

As I understand little about organic synthesis, I cannot say whether this part is good in the article. I just thought it was too long.

Response: In the revised version of the review, we enhanced its pharmacological part with mechanisms of action, a structural-active relationship analysis, and literature data.

Reviewer 4 Report

The presented manuscript is very good. The authors made the great job to collected, considered, and described to many literature data. The topic is actual and such paper will be interest to a broad audience, primarily, organic and medicinal chemists. Thus I recommend publishing this manuscript in IJMS, however after several corrections.

 1. Table of content and list of abbreviations are highly desired for such large review.

 2. Main text.

2.1. Many word should be Italic (cis, trans, bis, ortho, para, syn etc). The authors should check and correct the manuscript. Also “N-heterocycles”, “O-substituent”, and all other “atom” should be italicized.

2.2. I think that the type of metal center should be specified in some cases (for example, Rh(I) or Rh(III), Au(I) or Au(III), Cu(I) or Cu(II) catalysts). Also “iodine (III)” should be corrected as “iodine(III)” (line 246) “Cu (II)” should be corrected as  “Cu(II)” (line 577).

2.3. Line 153: “piperidins” should be corrected as "piperidines"

2.4. Line 654: should be “FeCl3*6H2O”

2.5. Line 627: “medicine. [136].”; should be “medicine [136].”

2.6. Lines 457–459: “In this approach, the diastereoselectivity of the final products is determined in the Mannich reaction, and the selectivity is maintained after the reduction of the intermediate” is not clear. Please rephrase.

2.7. Line 509: “enentioselective”

2.8. Line 724: “acetylene group source” should be corrected as “acetylene source”

2.9. In section 3 (Pharmacological applications of piperidine derivatives), compounds’ numbers (1, 2, 3 etc) should be bold.

2.10. Line 812: should be “carbonic anhydrase II (7), IX (8) and XII (9) inhibitors”

 3. Figures and Schemes

3.1. Schemes 1, 15, and 70–74 are Figures.

3.2. Room temperature should be mentioned as “RT” or “r.t.”; not “rt”.

3.3. In all schemes, Rx should be corrected as Rx (superscript should be used instead of subscript)

3.4. In all schemes, reaction times should be mentioned without dot: “1 h” instead of “1 h.” or “30 min” instead of “30 min.” In addition, it is recommended that the reaction time mentioned after reaction temperature.

3.5. “C6H4 is not “Ph”. For example, “CH2(4-CF3C6H4)” should be instead of “CH2(4-CF3-Ph)”. The authors should check and correct this in all Schemes

3.6. Scheme 38 is missed

3.7. Scheme 44: substituents “R2-4 are not clear

3.8. Scheme 58: should be “Isopropyl alcohol”, not “iso-propanol”

Author Response

We thank the reviewer for the comments!

The presented manuscript is very good. The authors made the great job to collected, considered, and described to many literature data. The topic is actual and such paper will be interest to a broad audience, primarily, organic and medicinal chemists. Thus I recommend publishing this manuscript in IJMS, however after several corrections.

  1. Table of content and list of abbreviations are highly desired for such large review.

Response: List of abbreviations was added as follows:

Substituent abbreviations: Ac – Acetyl; Ar – Aromatics; Bn – Benzyl; Boc – tert-Butyloxycarbonyl; Bu – Butyl; Cbz – Benzyloxycarbonyl; COD – Cyclooctadiene; coe – Cyclooctene; Cp – Cyclopentadienyl; dba – Dibenzylideneacetone; Et – Ethyl; Hbpin – Pinacolborane; hex – Hexyl; IMes –1,3-bis(2,4,6-trimethylphenyl)imidazol-2-ylidene; L – Ligand; Me – Methyl; Ms – Methanesulfonyl; Napht – Naphtyl; nbd – Norbornadiene; Ns – Nosyl; Ph – Phenyl; Phen – Phenanthroline; Pent – Pentyl; Piv – Pivaloyl; PMP – p-Methoxyphenyl; ppy – Phenylpyridine; Pr – Propyl; PS – Polystyrene; Py – Pyridine; Tf – Triflate; TPP – Tetraphenylporphine; Ts – toluenesulfonyl.

Reagent/Solvent abbreviations: DBAD – Dibenzyl azodicarboxylate; DBU – 1,8-Diazabicyclo[5.4.0]undec-7-ene; DIBAL-H – Diisobutylaluminium hydride; DCE – Dichloroethane; DCM – Dichloromethane; DMA – Dimethylacetamide; DMF – Dimethylformamide; DMPU – N,N′-Dimethylpropyleneurea; DMSO – Dimethyl sulfoxide; HATU – 2-(7-aza-1H-benzotriazole-1-yl)-1,1,3,3-tetramethyluronium hexafluorophosphate; HFIP – Hexafluoroisopropanol; KHDMS – Potassium bis(trimethylsilyl)amide; LiHDMS – Lithium bis(trimethylsilyl)amide; PEG - Polyethylene glycol; PIDA – Phenyliodine(III) diacetate; STAB - Sodium triacetoxyborohydride; TBAF – Tetra-n-butylammonium fluoride; TBHP – tert-Butyl hydroperoxide; TFA – Trifluoroacetic acid; TFE – Trifluoroethanol; THF – Tetrahydrofuran; TIPS – Triisopropyl silane; TMEDA – Tetramethylethylenediamine; TMS – Tetramethylsilane.

Term abbreviations: dr – Diastereomeric Ratio; ee – Enantiomeric Excess; er – Enantiomeric Ratio; HAT – Hydrogen Atom Transfer; IMAMR – Intramolecular Aza-Michael Reactions; MCR – Multicomponent Reaction; MOC – Memory of Chirality.

Medical abbreviations: 11β-HSD1 – 11β-Hydroxysteroid Dehydrogenase Type 1; Ache – Acetylcholinesterase; ALK – Anaplastic Lymphoma Kinase; ARPC – Androgen-Refractory Cancer Cell Lines; BACE-1 – Beta-secretase 1; Buche – Butyrylcholinesterase; CA – Carbonic Anhydrase; CGRP – Calcitonin Gene-Related Peptide Receptor; CNS – Central Nervous System; DNA – Deoxyribonucleic Acid; Ikkb – Iκb Kinase; M3R – M3 Muscarinic Acetylcholine Receptor; MAGL – Monoacylglycerol Lipase; MAO-B – Monoamine Oxidase B; MRSA – Methicillin-Resistant Staphylococcus Aureus; NF-κB – Nuclear factor kappa-light-chain-enhancer of activated B cells; ROS1 – C-Ros Oncogene 1; Trpv1 – Transient Receptor Potential Cation Channel Subfamily V Member 1; WHO – World Health Organization.

  1. Main text.

2.1. Many word should be Italic (cistransbisorthoparasyn etc). The authors should check and correct the manuscript. Also “N-heterocycles”, “O-substituent”, and all other “atom” should be italicized.

Response: Corrected accordingly

2.2. I think that the type of metal center should be specified in some cases (for example, Rh(I) or Rh(III), Au(I) or Au(III), Cu(I) or Cu(II) catalysts). Also “iodine (III)” should be corrected as “iodine(III)” (line 246) “Cu (II)” should be corrected as  “Cu(II)” (line 577).

Response: Corrected accordingly. Type of metal center was specified when possible.

2.3. Line 153: “piperidins” should be corrected as "piperidines"

Response: Corrected accordingly

2.4. Line 654: should be “FeCl3*6H2O”

Response: Corrected accordingly

2.5. Line 627: “medicine. [136].”; should be “medicine [136].”

Response: Corrected accordingly

2.6. Lines 457–459: “In this approach, the diastereoselectivity of the final products is determined in the Mannich reaction, and the selectivity is maintained after the reduction of the intermediate” is not clear. Please rephrase.

Response: Rephrased to “The diastereoselective Mannich reaction (first step) between functionalized acetals and imines is used to control the stereochemistry of piperidines, which is retained during reductive cyclization (second step).”

2.7. Line 509: “enentioselective”

Response: Corrected to “enantioselective”

2.8. Line 724: “acetylene group source” should be corrected as “acetylene source”

Response: Corrected accordingly

2.9. In section 3 (Pharmacological applications of piperidine derivatives), compounds’ numbers (1, 2, 3 etc) should be bold.

Response: Corrected accordingly

2.10. Line 812: should be “carbonic anhydrase II (7), IX (8) and XII (9) inhibitors”

Response: Corrected accordingly

  1. Figures and Schemes

3.1. Schemes 1, 15, and 70–74 are Figures.

Response: Corrected accordingly

3.2. Room temperature should be mentioned as “RT” or “r.t.”; not “rt”.

Response: Corrected accordingly

3.3. In all schemes, Rx should be corrected as Rx (superscript should be used instead of subscript)

Response: Corrected accordingly

3.4. In all schemes, reaction times should be mentioned without dot: “1 h” instead of “1 h.” or “30 min” instead of “30 min.” In addition, it is recommended that the reaction time mentioned after reaction temperature.

Response: Corrected accordingly

3.5. “C6H4 is not “Ph”. For example, “CH2(4-CF3C6H4)” should be instead of “CH2(4-CF3-Ph)”. The authors should check and correct this in all Schemes

Response: Corrected accordingly

3.6. Scheme 38 is missed

Response: Missing scheme is added

3.7. Scheme 44: substituents “R2-4 are not clear

Response: “R2-4 was corrected to “R2

3.8. Scheme 58: should be “Isopropyl alcohol”, not “iso-propanol”

Response: Corrected accordingly

Round 2

Reviewer 3 Report

Dear,

The new version has been corrected following the guidelines. Therefore, the manuscript can be considered approved as it is.